# Copper-catalyzed regio- and stereo-selective hydrosilylation of terminal allenes to access (E)-allylsilanes

Shaowei Chen[1], Xiaoqian He[2], Yi Jin[1], Yu Lan [2,3✉] & Xiao Shen [1✉]

Regioselectivity and stereoselectivity control in hydrosilylation of terminal allenes is challenging. Although the selective synthesis of vinylsilanes, branched allylsilanes or linear (Z)-allylsilanes have been achieved, transition-metal catalyzed hydrosilylation of terminal allenes to access (E)-allylsilane is difficult. Herein, we report a copper-catalyzed selective hydrosilylation reaction of terminal allenes to access (E)-allylsilanes under mild reaction conditions. The reaction shows broad substrate scope, representing an efficient method to prepare trisubstituted (E)-allylsilanes through hydrosilylation reaction of allenes and can also be applied in the synthesis of disubstituted (E)-allylsilanes. The mechanism study reveals that the E-selectivity is kinetically controlled by the catalyst but not by the thermodynamically isomerization of the (Z)-isomer.

[1] The Institute for Advanced Studies, Engineering Research Center of Organosilicon Compounds & Materials, Ministry of Education, Wuhan University, 430072 Wuhan, China. [2] School of Chemistry and Chemical Engineering, Chongqing Key Laboratory of Theoretical and Computational Chemistry, Chongqing University, 400030 Chongqing, PR China. [3] College of Chemistry and Molecular Engineering, Zhengzhou University, 450001 Zhengzhou, PR China. ✉email: lanyu@cqu.edu.cn; xiaoshen@whu.edu.cn

Organosilicon compounds are widely used in synthetic chemistry and material science[1]. As a step- and atom-economical approach to synthesize organosilicon compounds, transition-metal catalyzed hydrosilylation of unsaturated C–C bonds such as alkenes[2–10], alkynes[11–17] and dienes[18–24], has been extensively studied. However, the study of the selective hydrosilylation of allenes lags behind, probably because of the challenges associated with regioselectivity and stereoselectivity control[25]. As for terminal allenes, six possible isomers could be potentially generated in transition-metal catalyzed hydrosilylation reaction, due to the presence of two continuous orthogonal π bonds (Fig. 1a). Previous hydrosilylation of terminal allenes with Pd, Ni, Au, Ru or Al catalysis mainly occurred at the non-terminal C=C bonds, affording vinylsilanes as the major products[26–32]. The synthesis of branched allylsilanes with Pd or Ni catalysis have also been achieved, which also occurred at the non-terminal C=C bonds[27,28,31]. In 2016, Asako and Takai reported a molybdenum-catalyzed hydrosilylation at the terminal C=C bonds of allenes that yielded linear allylic silanes in moderate to good Z-selectivities[33]. Seminal work in the transition-metal catalyzed synthesis of (Z)-allylsilanes were then reported by Ma and Huang groups and Ge group with cobalt catalysis[34,35]. However, there is still one challenge remained in transition-metal catalyzed hydrosilylation of terminal allenes, that is, the regio- and stereoselective generation of (E)-allylsilanes. The only report to prepare (E)-allylsilanes via hydrosilylation of allenes was Yao's radical-based approach, but the reaction is limited to super reactive (TMS)$_3$SiH and monoalkyl substituted allenes[36]. There are several other methods to access di-substituted (E)-allylsilanes[37–39], but no general catalytic hydrosilylation methods to prepare tri-substituted (E)-allylsilanes is availabe. To the best of our knowledge, transition metal catalyzed hydrosilylation of terminal allenes to obtain both trisubstituted and disubstituted (E)-allylsilanes has not been well studied.

It is worth to mention that copper is an earth abundant transition metal, which makes it an ideal candidate to develop transformations in sustainable chemistry. However, compared to relatively more extensively studied Ni, Co, Fe-catalyzed hydrosilylation reactions[2,40–42], copper catalysis has been rarely used in hydrosilylation of unsaturated carbon carbon bonds[24,43–49]. Herein, we report a copper-catalyzed hydrosilylation of allenes which affords linear (E)-allylsilanes with excellent regio- and stereoselectivity. The reaction shows broad substrate scope and is amenable to synthesize both di-substituted and tri-substituted (E)-allylsilanes. The mechanism study reveals that the E-selectivity is kinetically controlled by the catalyst but not by the thermodynamically isomerization of the (Z)-isomer.

## Results

**Evaluation of reaction conditions.** We commenced the study of Cu-catalyzed hydrosilylation of terminal allenes by evaluating the reaction conditions with buta-2,3-dien-2-ylbenzene **1a** and Ph$_2$SiH$_2$ **2a** as the model substrates. When 5 mol% of Cu(OAc)$_2$ was used as catalyst, the reaction in THF at room temperature afforded the desired allylsilane in 7% yield with 57:43 E/Z (Table 1, entry 1). We then screened various ligands to increase the efficiency of the reaction (entries 2–7). When monophosphine ligand PPh$_3$ was used as a ligand, the E/Z ratio increased to 92:8, but only 5% yield was obtained (entry 2). Several bisphosphine ligands were then found to be better than PPh$_3$. When Xantphos is used, the yield increased to 69%, and the E-selectivity increased to 99% (entry 3). Compared with Xantphos, other bisphosphine ligands, such as DPPE, DPPB, DPPM, DPPBZ, could only afford the product in 15–25% yield (92–99% E selectivity, entries 4–7). Further study revealed that increasing the amount of **2a** to 1.2 equivalent could improve the yield to 74% (entry 8). Then we screened the molecular ratio of Cu(OAc)$_2$/Xantphos, and the best result was obtained when the amount of Xantphos was increased to 7.5%, while 5 mmol% Cu(OAc)$_2$ was used, and the yield increased to 90% (entry 10). In all cases, less than 1% yield was observed for vinylsilane **3a'**, branched allylsilane **3a"** or

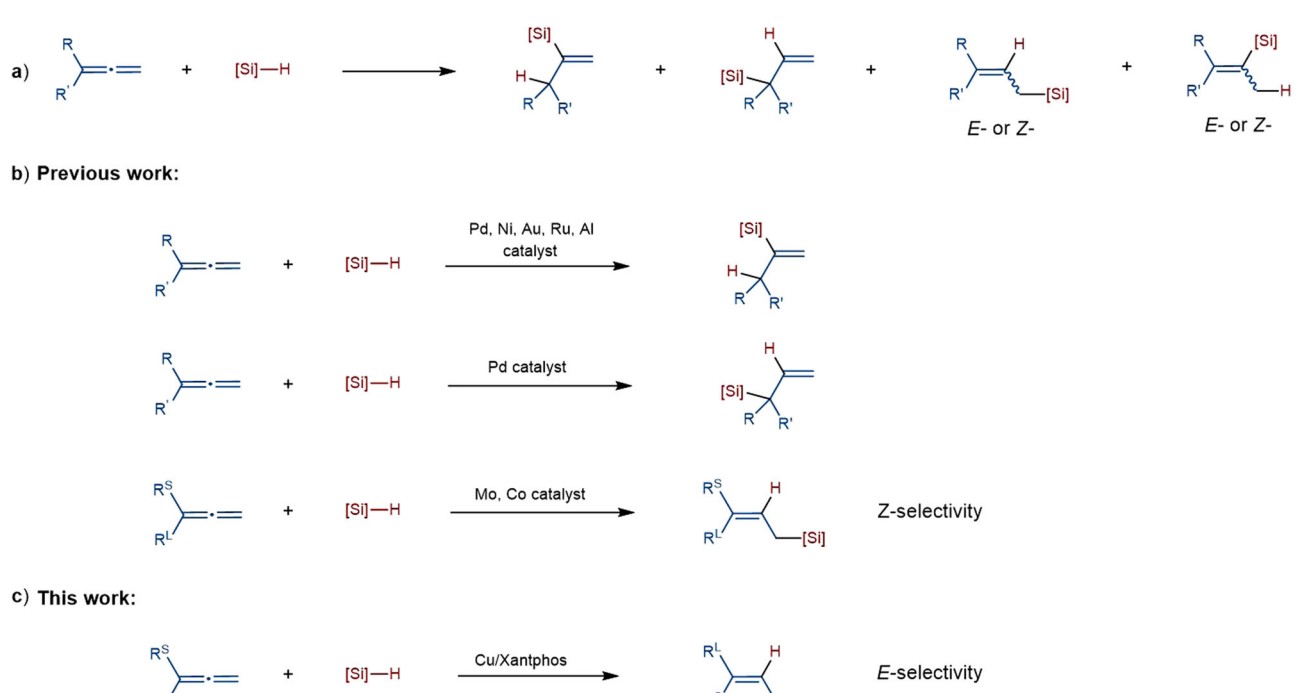

**Fig. 1 Hydrosilylation of terminal allenes. a** Challenges in the control of regio- and stereo-selectivity. **b** Previous work on the transition-metal-catalyzed hydrosilylation of terminal allenes mainly afforded vinyl silanes, branched allylsilanes and (Z)-allylsilanes as the products. **c** This work: we report a copper-catalyzed selective hydrosilylation of terminal allenes to access exclusively (E)-allylsilanes.

**Table 1 Reaction optimization.**

| entry | ligand | conv. of 1a | yield of 3a | E/Z of 3a |
|---|---|---|---|---|
| 1 | none | 34% | 7% | 57:43 |
| 2[b] | PPh₃ | 16% | 5% | 92:8 |
| 3 | Xantphos | 100% | 69% | 99:1 |
| 4 | DPPE | 33% | 15% | 98:2 |
| 5 | DPPB | 26% | 16% | 99:1 |
| 6 | DPPM | 58% | 25% | 92:8 |
| 7 | DPPBZ | 36% | 14% | 98:2 |
| 8[c] | Xantphos | 100% | 74% | 99:1 |
| 9[d] | Xantphos | 100% | 84% | 99:1 |
| 10[e] | Xantphos | 100% | 90% | 99:1 |
| 11[f] | Xantphos | 100% | 88% | 99:1 |

Xantphos  DPPE  DPPB  DPPM  DPPBZ

[a]The mixture of **1a** (0.2 mmol), Cu(OAc)₂ (5 mmol%, 0.01 mmol), ligand (5 mmol%, 0.01 mmol), **2a** (0.2 mmol) in THF (0.4 mL) was stirred at rt under N₂ for 12 h. Yield of **3a** was determined by ¹H NMR using BrCH₂CH₂Br as an internal standard, Z/E ratio was determined by GC-MS analysis of tye unpurified reaction mixtures.
[b]PPh₃ (10 mmol%, 0.02 mmol).
[c]**2a** (0.24 mmol, 1.2 equiv.).
[d]**2a** (0.24 mmol, 1.2 equiv.), Xantphos (6 mmol%, 0.012 mmol).
[e]**2a** (0.24 mmol, 1.2 equiv.), Xantphos (7.5 mmol%, 0.015 mmol).
[f]**2a** (0.24 mmol, 1.2 equiv.), Xantphos (10 mmol%, 0.02 mmol).

vinylsilane **3a'''**, indicating the excellent regio- and stereo-selectivity control under the copper catalyzed conditions.

**Scope of the reaction between allene 1 and silane 2.** Using the optimized conditions, we first examined the scope of 1,1-disubsitituted allenes. These results are summarized in Fig. 2. A wide range of 1,1-disubstituted allenes with various subsituents on the aromatic rings such as Me, Et, t-Bu, F, Cl, Br, CF₃, OMe, OCF₃, ester and MeS were suitable substrates, affording the desired (E)-allylsilanes **3a–3v** in 50–90% yield with good to excellent E-selectivities (96:4–99:1 E/Z). Additionally, electron-rich thiophenyl group could be tolerated (**3n**, 50% yield, 97:3 E/Z). (E)-allylsilanes which contain naphthyl, biphenyl groups have also been successfully made (**3o**, 84% yield, 99:1 E/Z; **3p**, 83% yield, 99:1 E/Z). If the aryl substituent in the disubstituted allene is replaced with cyclohexyl (**3w**), a yield of 88% can be obtained, but the E/Z ratio will decrease to 88:12. Then we tested several diarylsilanes with **1a** as the reaction partner, and the corresponding products were obtained in good yields and excellent selectivities (**3x**, 70% yield, 98:2 E/Z; **3y**, 70% yield, 99:1 E/Z; **3z**, 81% yield, 99:1 E/Z; **3aa**, 84% yield, 99:1 E/Z; **3ab**, 83% yield, 99:1 E/Z). Phenylsilane was also suitable reagent for the selective hydrosilylation reaction, and compound **3ac** was synthesized in 66% yield with 98:2 E/Z under slightly modified reaction conditions. However, reducing the steric difference between two substituents of 1,1-disubstituted allenes decreased the E/Z selectivity of the hydrosilylation reaction between (4-methylpenta-1,2-dien-3-yl)benzene and Ph₂SiH₂ (**3ad**, 79% yield, 66:34 E/Z).

Monosubstituted allenes also reacted with **2a** smoothly under the slightly modified reaction conditions (Fig. 3). Because of the higher reactivity of 1,2-disubstituted alkenes than the tri-substituted alkenes, the subsequent hydrosilylation of the

(E)-allylsilane product was observed, in the presence of excess amount of Ph₂SiH₂. Therefore, the amount of silane was reduced to 1.1 equivalent from 1.2 equivalent and the reaction time was shortened from 12 h to 6 h to get high yield of mono-hydrosilyaltion product. Allenes substituted with electron-donating groups such as Me, Et, t-Bu, OMe, reacted well with **2a** to afford (E)-allylsilanes **5a–5g** in 70–88% yield with excellent E-selectivity (99:1 E/Z). Allenes substituted with halogen atoms (F, Cl, Br) were suitable substrates, affording the desired (E)-allylsilanes **5h-5i** in 66–82% yield with 98:2–99:1 E/Z. (E)-allylsilanes which contain naphthyl, biphenyl groups have also been successfully prepared (**5m**, 87% yield, 99:1 E/Z; **5o**, 71% yield, 99:1 E/Z). The reaction of **2a** with fluorine-substituted silanes also performed well, affording (E)-allylsilanes efficiently (**5p**, 74% yield, 99:1 E/Z; **5q**, 83% yield, 99:1 E/Z; **5r**, 81% yield, 99:1 E/Z). Alkyl allene afforded compound **5t** in 58% yield with 89:11 E/Z. When phenylsilane was used as the reagent, allylsilane **5u** was isolated in 72% yield with 98:2 E/Z. Benzyl(hexyl)silane and methyl(phenyl)silane were also tested in the hydrosilylation reactions with allene **1a**, and compounds **5v** and **5w** were afforded in 53% yield, 98:2 E/Z and 70% yield, 96:4 E/Z, respectively.

**Synthetic transformations of compound 3a.** To test synthetic potential of this reaction, we scaled up the model reaction to 10 mmol scale, and allylsilane **3a** was isolated in 83% yield (2.6 g) with 99:1 E:Z (Fig. 4a). Then we explored the conversion of Si-H bond of **3a** with the retention of the carbon carbon double bond (Fig. 4b). Taking advantage of the E-configuration of compound **3a**, we achieved the intramolecular dehydrogenative C-Si bond formation via Ir catalyzed C-H activation, affording the five-membered silicon-containing compound **6** in 60% yield. The

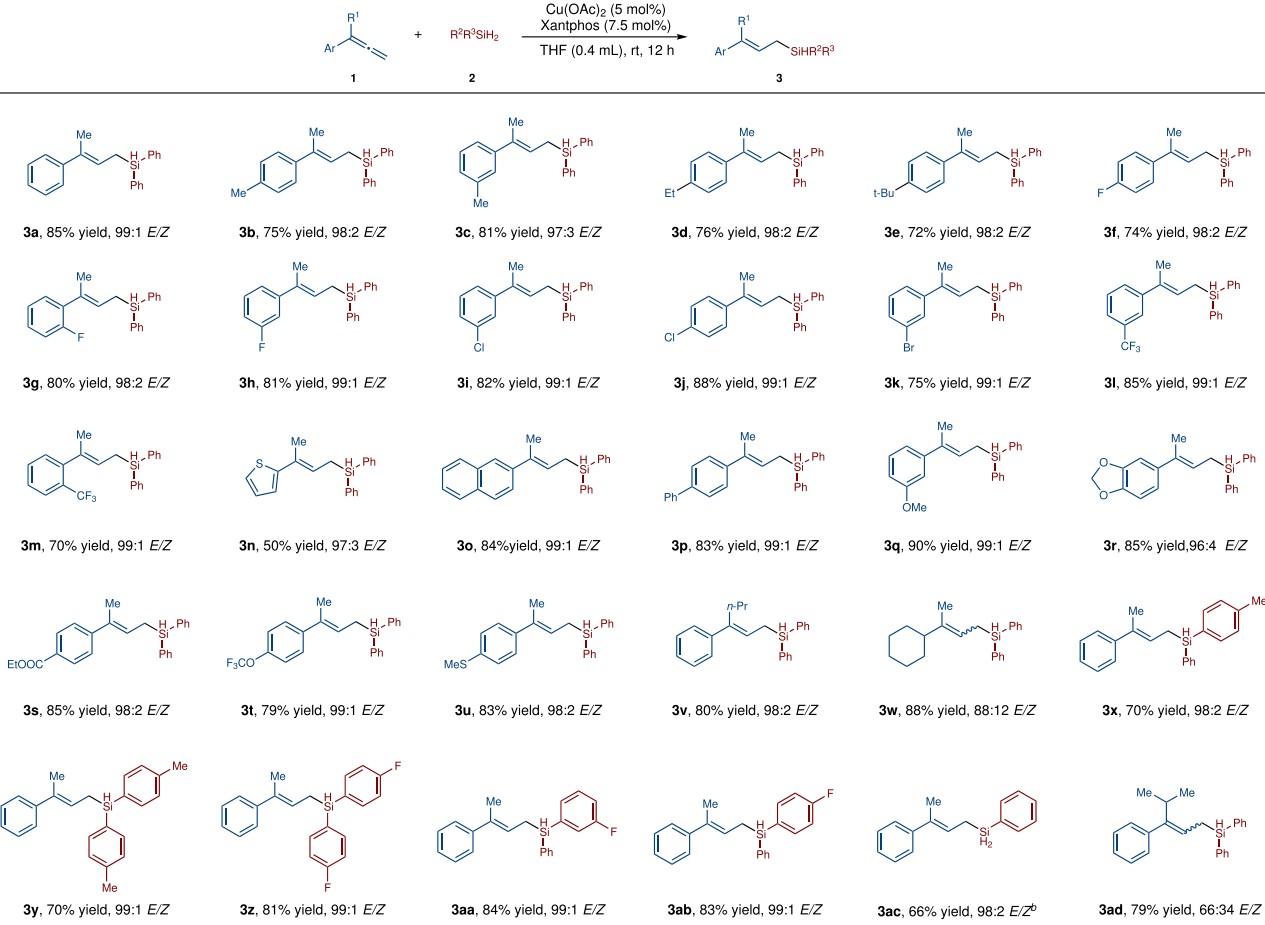

**Fig. 2 Scope of 1,1-disubstituted allenes.** Otherwise noted, the reaction was performed under the following conditions:[a] allene (0.20 mmol), $R^2R^3SiH_2$ (0.24 mmol), $Cu(OAc)_2$ (0.01 mmol, 5 mol%), Xantphos (0.015 mmol, 7.5 mol%), THF (0.4 mL), rt, 12 h; the yield refers to isolated yield; E/Z ratio was determined with GC-MS analysis or HNMR of unpurified reaction mixture[b]. Allene (0.20 mmol), $PhSiH_3$ (0.4 mmol), $Cu(OAc)_2$ (0.01 mmol, 5 mol%), Xantphos (0.015 mmol, 7.5 mol%), THF (0.4 mL), 30 °C, 3 h.

reaction of compound **3a** with MeOH in the presence of a N-heterocyclic carbene catalyst afforded siloxane **7** in 85% yield. Treatment of **3a** with MeLi at room temperature gave allylsilane **8** in 80% yield. Allylic alcohols are synthetically valuable intermediates in various organic transformations[50,51]. Oxidation of the (E)-allylsilanes with $H_2O_2$ under basic conditions afford allylic alcohol **9** in 91% yield. In all of these reactions, the configurations of the double bond did not change. Moreover, Allyl silanol **10** could be obtained in 90% yield under the Pd-catalyzed conditions (Fig. 4c). Compound **11** was then synthesized in 77% yield through the Hiyama-Denmark cross-coupling reaction between silanol **10** and PhI (Fig. 4c).

## Discussion

In order to understand the mechanism of the copper-catalyzed hydrosilylation of terminal allenes, we conducted a deuterium-labeling reaction between **1a** and $Ph_2SiD_2$. This reaction afforded the corresponding (E)-allylsilane with >99% D-incorporation, indicating that the hydrogen atom was from the silane, but not from the solvent (Fig. 5a). The KIE study indicated that Si-H bond might be involved in the rate-determining step (Fig. 5b). (Z)-allylsilanes are thermodynamically less stable than (E)-allylsilanes. We wonder whether the high E-selectivity of our reaction was resulted from catalyst control or the thermodynamically isomerization of the less stable (Z)-isomer to the more stable (E)-isomer? Firstly, we tested the reaction of (Z)-**3a** under the

standard conditions in the presence of 1 equivalent of $Ph_2SiH_2$, and we found that there was no Z/E-isomerization (Fig. 5c). Moreover, compound **5h** (91:9 E/Z) did not undergo Z/E-isomerization either (Fig. 5d). These experimental results strongly support that the E-stereoselectivity of our reaction is kinetically controlled. It is known that CuH intermediates could be generated from the reaction of copper salts and silanes and CuH could add to olefins to generate organocopper intermediates[43–49]. For the reaction of allenes with CuH, both terminal and internal double bonds of **1a** could participate in the hydrocupration[47,48]. In order to further understand the origin of the stereo- and regioselectivity of our reaction, density functional theory (DFT) calculations have been performed (Fig. 6, Supplementary Information and Supplementary Data 1)[52,53]. The calculation results showed that intermediate **16** is more stable than intermediates **17** and **18**. Although the energy barrier to generate **17** (**14-ts**, $\Delta G^{\ddagger} = 17.1$ kcal/mol) is lower than those to generate **16** (**13-ts**, $\Delta G^{\ddagger} = 17.8$ kcal/mol) and **18** (**15-ts**, $\Delta G^{\ddagger} = 18.4$ kcal/mol), the 1,3-Cu transfer among these intermediates are rather easy with energy barriers of about 6 kcal/mol. The subsequent σ-bond metathesis step of the $Ph_2SiH_2$ **2a** and allylic copper intermediates **16**, **17** and **18** could proceed via the four-membered cyclic transition states. Among the three transition states, **21-ts** ($\Delta G^{\ddagger} = 23.7$ kcal/mol) which leads to the formation of (E)-**3a**, is found to be the most favorable one (**22-ts**, $\Delta G^{\ddagger} = 25.0$ kcal/mol; **23-ts**, $\Delta G^{\ddagger} = 25.4$ kcal/mol). The DFT calculation results are consistent to our experimental data, that is, (E)-allylsilanes were

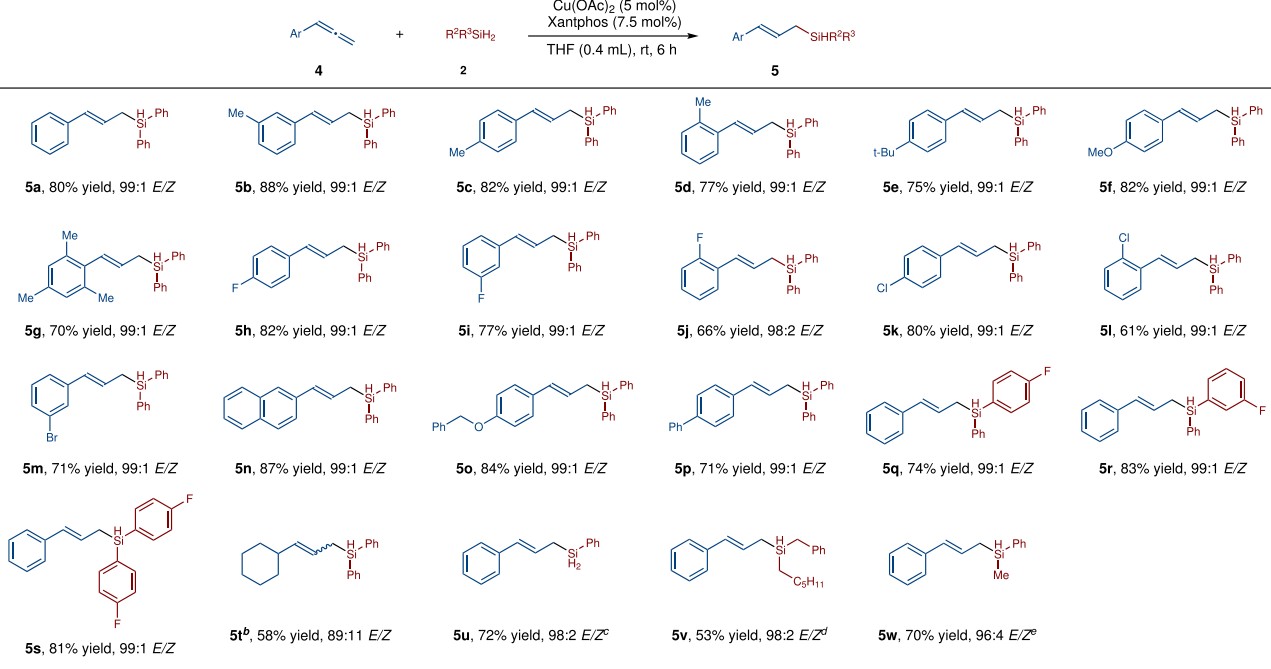

**Fig. 3 Scope of monosubstituted allenes.** Otherwise noted, the reaction was performed under the following conditions:[a] allene (0.20 mmol), R2R3SiH2 (0.22 mmol), Cu(OAc)2 (0.01 mmol, 5 mol%), Xantphos (0.015 mmol, 7.5 mol%), THF (0.4 mL), rt, 6 h; the yield refers to the isolated yield; Z/E ratio was determined by GC-MS or HNMR analysis of the unpurified reaction mixtures[b]. Allene (0.40 mmol), Ph2SiH2 (0.2 mmol), The yield of **5t** was determined by HNMR[c]. Allene (0.20 mmol), PhSiH3 (0.4 mmol), Cu(OAc)2 (0.005 mmol, 2.5 mol%), Xantphos (0.0075 mmol, 3.75 mol%), THF (0.4 mL), rt, 30 min[d]. Allene (0.20 mmol), silane (0.4 mmol), Cu(OAc)2 (0.02 mmol, 10 mol%), Xantphos (0.03 mmol, 15 mol%), THF (0.4 mL), 30 ºC, 12 h[e]. Allene (0.20 mmol), PhMeSiH2 (0.4 mmol), Cu(OAc)2 (0.01 mmol, 5 mol%), Xantphos (0.015 mmol, 7.5 mol%), THF (0.4 mL), 30 ºC, 3 h.

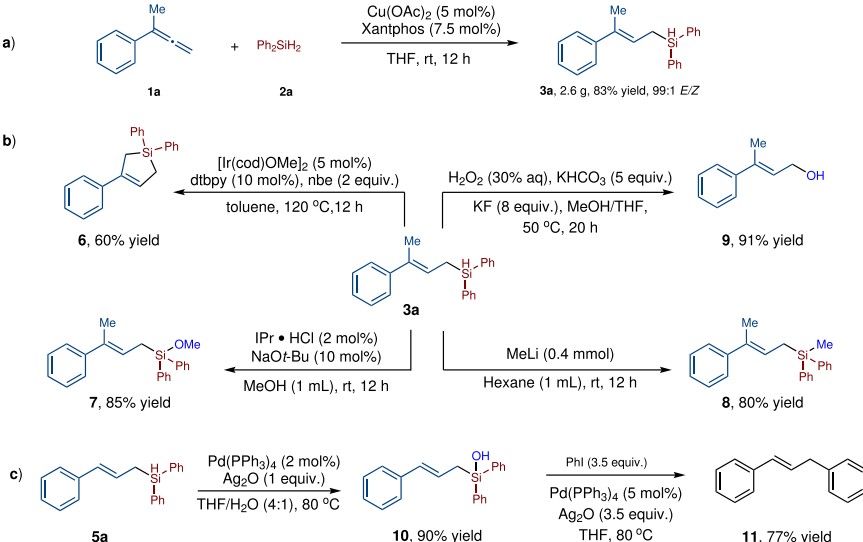

**Fig. 4 Gram scale reaction and downstream transformations. a** The reaction on 10 mmol performed well. **b** Synthetic transformations of compound **3a**. **c** Synthetic transformations of compound **5a**.

observed as the major products. According to DFT calculations, σ-bond metathesis step are the reaction rate-limiting step and the stereoselectivity determining step, which is consistent to result of the KIE study.

Independent gradient model (IGM) analysis of the transition states **21-ts**, **22-ts** and **23-ts** was also performed[54,55]. As shown in Fig. 6c, **21-ts** is 1.3 and 1.7 kcal/mol more favorable than **22-ts** and **23-ts**, which is consistent with the stereo- and regioselectivity observed in the experiment. The leading factor that differentiates the three competing transition states is likely to be π-π interaction between the

phenyl group of Xantphos and the phenyl group of the allene. This favorable π-π interaction stabilizes **21-ts** ($\Delta E_{\pi-\pi} = -2.1$ kcal/mol) by 0.7 and 2.0 kcal/mol relative to **22-ts** ($\Delta E_{\pi-\pi} = -1.4$ kcal/mol) and **23-ts** ($\Delta E_{\pi-\pi} = -0.1$ kcal/mol) based on the calculations of interacting fragments. The green oval represents the presence of interactions between the highlighted fragments. Therefore, our calculations indicate noncovalent π–π interaction is the determinant of stereo- and regioselectivity.

In summary, we have developed a copper-catalyzed regio- and stereo-selective hydrosilylation of allenes to access (E)-allylsilanes.

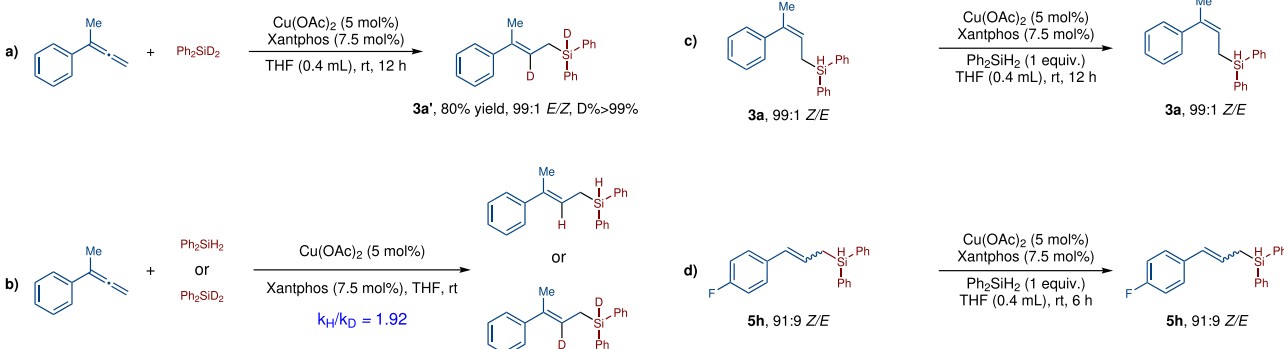

**Fig. 5 Mechanism study. a** A deuterium-labeling reaction indicate that the hydrogen comes from the silane. **b** KIE study suggested that the rate-determining step might involve in the reaction of Si-H bond. **c** No isomerization of **Z-3a** to **E-3a** suggest that the selectivity is kinetically controlled. **d** No change of the *Z/E* ratio of **5h** suggest that our Cu-catalyzed reaction is kinetically controlled.

**Fig. 6 Computational study. a** DFT calculations for the Cu-catalyzed hydrocupration of allene **1a**. **b** Gibbs free energies required for the σ-bond metathesis step of Ph$_2$SiH$_2$ **2a** and three allylic copper intermediates **16**, **17** and **18**. **c** DFT calculations on the π-π interaction between the phenyl group of Xantphos and the phenyl group of the allene. Calculations were performed using Gaussian 16 at the M06-L/SDD-6-311 + G(d,p)/SMD(THF)//B3LYP-D3(BJ)/LANL2DZ-6-31G(d) level of theory and the values are shown in kcal/mol.

A wide range of 1,1-disubstituted and monosubstituted terminal allenes reacted to afford the corresponding (E)-allylsilanes in good yields. The reaction conditions are simple and mild, and the product can be prepared in grams, which proves the practicability of this reaction. The mechanism study reveals that the E-selectivity is kinetically controlled by the catalyst but not by the thermodynamically isomerization of the (Z)-isomer.

## Methods

**General procedure for copper-catalyzed allene hydrosilylation.** In a glovebox, to an oven-dried screw-capped 4 ml glass vial equipped with a magnetic stir bar was added $Cu(OAc)_2$ (0.01 mmol, 5 mol%), Xantphos (0.015 mmol, 7.5 mol%), THF (0.4 mL). The mixture was stirred for 15 min. Then terminal allene (0.2 mmol, 1.0 equiv.) and silane (0.24 mmol, 1.2 equiv. or 0.22 mmol, 1.1 equiv.) were added and the mixture was stirred at room temperature for 12 h (Figs. 2) or 6h (Fig. 3). The solvent was removed under vacuum and the residue was purified by column chromatography to afford the corresponding product. See section 1.3 and 1.4 in the Supplementary Information for more details.

## Data availability

The authors declare that all other data supporting the findings of this study are available within the article, Supplementary Information and Supplementary Data, and also are available from the corresponding authors on request. Supplementary Data File 1 contains the cartesian coordinates and energies of optimized structures.

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

## Acknowledgements

We are grateful to NSFC (21901191, XS; 21822303, YL), Fundamental Research Funds for the Central Universities (XS) and Wuhan University (XS) for financial support. The numerical calculations in this paper have been done on the supercomputing system in the Supercomputing Center of Wuhan University.

## Author contributions

X.S. directed the project and composed the manuscript with revisions provided by the other authors. S.C. developed the catalytic method and performed the mechanism study and synthetic applications. S.C. and Y.J. investigated the substrate scope. X.H. performed the DFT calculations. Y.L. directed the calculation. All the authors were involved the analysis of results and discussions of the project.

## Competing interests

The authors declare no competing interests.
