## [Peer Review File · Nature Communications]

Copper-catalyzed regio- and stereo-selective hydrosilylation of terminal allenes to access (E)-allylsilanesREVIEWER COMMENTS

Reviewer #1 (Remarks to the Author):

Hydrosilylation is one of the most useful and fundamental reactions in organic synthesis. Shen and co-workers reported a copper-catalyzed regio- and stereoselective hydrosilylation of terminal allenes to afford E-allylsilanes. It is an interesting result. For the aryl allenes, high stereoselectivity has been observed. However, the stereoselectivity is decreased to 88/12 for the 1,1-dialkyl substituted allenes, and no example for mono-substituted alkyl allene. Although various diaryl hydrosilanes have been used, it is better to know the results by using phenyl hydrosilane or alkyl silane. Due to the previous studies on the synthesis of Z-allylsilanes, it is better to know the detail of the difference on stereoselectivity. The manuscript could be considered after addressing the above concern.

Reviewer #2 (Remarks to the Author):

Copper-catalyzed regio- and stereo-selective hydrosilylation of terminal allenes to access (E)-allylsilanes

Key results

Shen and co-workers report a method to access (E)-allylsilanes by copper catalysed hydrosilylation of terminal allenes. Reaction occurs under mild conditions, using copper (an Earth abundant element with low toxicity) as catalyst and gives access to a broad range of di-substituted and tri-substituted allyl silanes with excellent regio- and stereoselectivity. This reaction represents an easy and useful method to prepare (E)-allylsilanes. The synthetic potential of this reaction is demonstrated by scaling-up the reaction to gram scale and conducting synthetic transformations of the product. Mechanistic investigations confirm that E-selectivity is not controlled by the thermodynamically isomerisation of the less stable Z-isomer.

Validity

Most results of the study are valid. I do, however, have some concern over some of the claims the authors are making, and the level of experimental detail required to support these.

- Synthetic potential of products: Authors report synthetic transformations of compound 3a

which involve conversion of Si-H bond and oxidation to give the corresponding allylic alcohol with retention of E configuration. However, no more complex molecules are synthesised from this product and although I agree allylic alcohols are useful molecules there are simpler methods to prepare similar compounds. If the authors are to make this claim they need to provide clear evidence that these molecules are synthetically useful for example in cross-coupling reactions.

- Mechanistic understanding: There are no comments about regioselectivity. Regarding stereoselectivity, authors provide strong evidence supporting that (Z) isomer is not obtained from isomerisation of the less stable (E) isomer. However, it is confusing the conclusion of stereoselectivity kinetically controlled by the catalyst when they also suggest that stereoselectivity is probably determined by the thermodynamic stability of the (E)-allylcopper intermediates. Is the formation of allylcopper intermediate the rate limiting step? Is the formation of (Z)-allylcopper reversible and (E)-isomer irreversible? In my opinion, without a more detailed understanding of the mechanism of these reactions, it may be a challenge to draw valid conclusions.

Significance

The paper makes an important addition to the field of hydrosilylation of terminal allenes. Authors report a simple method with a broad substrate scope to obtain both di- and trisubstituted allylsilanes with excellent regio- and stereoselectivity. I think these findings are of interest to a broad chemist community.

Data and methodology and Analytical Approach

In general, the data and findings have been presented to a very high standard. There are however several points the authors need to consider.

- ¹⁹F NMRs are not reported for any of the fluorinated products.
- ¹H NMR characterisation for compound 5t is not clear. Signals integrating 0.33H or 1.4H (for example) don't make sense. Resonances corresponding to E- and Z- isomers should be listed separately when possible.
- ¹H NMR characterisation of 3a' (deuterated 3a) should be reported. If possible, include ²H NMR data.

Suggested improvements

- Synthetical utility of the products could be proved for example by conducting coupling reactions.
- Study of the KIE of the reaction could shed light into the mechanisms of the hydrosilylation reaction.

Clarity and context

In general, the work is presented clearly. However, as stated above the authors need perhaps be a bit more precise on how stereoselectivity is controlled.

Aspects of discussion of the prior literature (e.g. 'On the basis of the above experimental results and previous reports about Cu-H chemistry') may also be a little hard to follow for those without a detailed knowledge of this field.

References

In general, references are appropriate. However, authors should consider include references about 'previous reports about Cu-H chemistry' in which the proposed mechanism is based (see above). A reference supporting synthetic value of allylic alcohols could be also added.

Reviewer #3 (Remarks to the Author):

This interesting manuscript describes the facile method for the synthesis of functionalized (E)-allylsilanes via copper-catalyzed regio- and stereoselective hydrosilylation of allenes.

There are not many examples of selective hydrosilylation of allenes because the reaction can potentially lead to many isomers.

The authors cite several examples of synthetic methods for obtaining (E)-allylsilanes via hydrosilylation of allenes but these examples are not without their drawbacks and limitations. Therefore, the reviewed work seems to be very interesting.

Unfortunately, a very similar paper on hydrosilylation of allenes has been published recently (<https://doi.org/10.1021/jacs.2c00260>).

The copper catalyst (Copper(I) thiophene-2-carboxylate) also proved to be an active and selective catalyst for the synthesis of functionalized (E)-allylsilanes in the presence of Xantphos. The investigations are very similar, and therefore, I do not see the possibility to publish this manuscript in this journal.

However, some points should be addressed.

- Figure 2 and 3 are incorrect Ph_2SiH_2 instead of PhSiH_2 and $\text{Ph}(\text{Ar})\text{SiH}_2$ instead of PhSiH_2
- PhSiH_3 and PhMeSiH_2 should also be tested
- Are compounds 3a-3z and 3aa, 3ab the isomers shown in fig.2? There is a lack of evidence. This is not apparent from the NMR spectra. Far interaction spectra or X-ray structures are missing.
- for the compounds from fig.3 only for a few examples ^1H NMR double bond coupling constants are shown.

RESPONSE TO REVIEWER COMMENTS

Reviewer #1 (Remarks to the Author):

Hydrosilylation is one of the most useful and fundamental reactions in organic synthesis. Shen and co-workers reported a copper-catalyzed regio- and stereoselective hydrosilylation of terminal allenes to afford *E*-allylsilanes. It is an interesting result. For the aryl allenes, high stereoselectivity has been observed.

Our response: Thanks for the positive comments.

However, the stereoselectivity is decreased to 88/12 for the 1,1-dialkyl substituted allenes, and no example for mono-substituted alkyl allene.

Our response: Thanks for the comments. We have tested a mono-substituted alkyl allene, and 58% yield with 89:11 *E/Z* was afforded for compound **5t** (Fig 3).

5t, 58% yield, 89:11 *E/Z*

Although various diaryl hydrosilanes have been used, it is better to know the results by using phenyl hydrosilane or alkyl silane.

Our response: Thanks for the kind suggestions. Phenyl hydrosilane and alkyl silane are also suitable reagents for our reaction, and the data of compound **3ac**, **5u**, **5v** and **5w** have been added in the revised Fig 2 and Fig 3.

3ac, 66% yield, 98:2 *E/Z*

5u, 72% yield, 98:2 *E/Z*

5v, 53% yield, 98:2 *E/Z*

5w, 70% yield, 96:4 *E/Z*

Due to the previous studies on the synthesis of *Z*-allylsilanes, it is better to know the detail of the difference on stereoselectivity. The manuscript could be considered after addressing the above concern.

Our response: Thanks for the kind suggestions. In order to understand the origin of the stereoselectivity of our reaction, density functional theory (DFT) calculations has been performed (Fig R1). The calculation results showed that intermediate **16** is more stable than intermediates **17** and **18**. Although the energy barrier to generate **17** is lower than those to generate **16** and **18**, the 1,3-Cu transfer among these intermediates are rather easy with energy barriers of about 6 kcal/mol. The subsequent σ -bond metathesis step of the Ph_2SiH_2 **2a** and allylic copper intermediates **16**, **17** and **18** could proceed via the four-membered cyclic transition states. Among the three transition states, **21-ts** (23.7 kcal/mol) which leads to the formation of (*E*)-**3a**, is found to be the most favorable one. The DFT calculation results are consistent to our experimental data, that is, (*E*)-allylsilanes were observed as the major products. According to DFT calculations, σ -bond metathesis step are the reaction rate-limiting step and the stereoselectivity determining step, which is consistent to result of

the KIE study ($k_H/k_D = 1.92$, Fig R2). We have added these data in the revised manuscript and supplementary information. The original catalytic cycle has been deleted and the data of compound **3ad** has been added in Fig 2 of the revised manuscript.

Fig. R1 Computational study. **a**) DFT calculations for the Cu-catalyzed hydrocupration of allene **1a**. **b**) Gibbs free energies required for the σ -bond metathesis step of Ph_2SiH_2 **2a** and three allylic copper intermediates **16**, **17** and **18**. Calculations were performed using Gaussian 16 at the M06-L/SDD-6-311++G(d,p)/SMD(THF)//B3LYP-D3(BJ)/LANL2DZ-6-31G(d) level of theory.

Fig. R2 KIE study

Reviewer #2 (Remarks to the Author):

Key results

Shen and co-workers report a method to access (*E*)-allylsilanes by copper catalysed hydrosilylation of terminal allenes. Reaction occurs under mild conditions, using copper (an Earth abundant element with low toxicity) as catalyst and gives access to a broad range of di-substituted and tri-substituted allyl silanes with excellent regio- and stereoselectivity. This reaction represents an easy and useful method to prepare (*E*)-allylsilanes. The synthetic potential of this reaction is demonstrated by scaling up the reaction to gram scale and conducting synthetic transformations of the product. Mechanistic investigations confirm that *E*-selectivity is not controlled by the thermodynamically isomerisation of the less stable *Z*-isomer.

Our response: Thanks for the positive comments.

Validity

Most results of the study are valid. I do, however, have some concern over some of the claims the authors are making, and the level of experimental detail required to support these.

-Synthetic potential of products: Authors report synthetic transformations of compound **3a** which involve conversion of Si-H bond and oxidation to give the corresponding allylic alcohol with retention of *E* configuration. However, no more complex molecules are synthesised from this product and although I agree allylic alcohols are useful molecules there are simpler methods to prepare similar compounds. If the authors are to make this claim they need to provide clear evidence that these molecules are synthetically useful for example in cross-coupling reactions.

Our response: Thanks for the comments and kind suggestions. Taking advantage of the *E*-configuration of compound **3a**, we achieved the intramolecular dehydrogenative C-Si bond formation via Ir catalyzed C-H activation, affording the five-membered silicon-containing compound **6** in 60% yield. Moreover, Allyl silanol **10** could be obtained in 90% yield under the Pd-catalyzed conditions. Compound **11** was then synthesized in 77% yield through the Hiyama-Denmark cross-coupling reaction between silanol **10** and PhI. These data have been added in the revised manuscript and supplementary information.

-Mechanistic understanding: There are no comments about regioselectivity. Regarding stereoselectivity, authors provide strong evidence supporting that (*Z*) isomer is not obtained from isomerisation of the less stable (*E*) isomer. However, it is confusing the conclusion of stereoselectivity kinetically controlled by the catalyst when they also suggest that stereoselectivity is probably determined by the thermodynamic stability of the (*E*)-allylcopper intermediates. Is the formation of allylcopper intermediate the rate limiting step? Is the formation of (*Z*)-allylcopper

reversible and (E)-isomer irreversible? In my opinion, without a more detailed understanding of the mechanism of these reactions, it may be challenge to draw valid conclusions.

Our response: Thanks for the helpful comments. To clarify the reaction mechanism, density functional theory (DFT) calculations have performed on the reaction of allene **1a** and Ph_2SiH_2 **2a**.

(1) Our calculation indicates that the hydrocupration step favors terminal C=C bond (**13-ts**, 17.8 kcal/mol; **14-ts**, 17.1 kcal/mol) over internal C=C bond (**15-ts**, 18.4 kcal/mol) in allene **1a**.

(2) Three allylic copper intermediates **16**, **17** and **18** could be provided from the hydrocupration, among which the kinetically less favorable *E*-isomer of the terminal allylic copper intermediate **16** is the thermodynamically most stable. Further calculation reveals that intermediates **16**, **17** and **18** may isomerize to each other via 1,3-Cu transfer, and these interconversions are rather easy with barriers of around 6 kcal/mol (Fig. R1). The subsequent σ -bond metathesis step of the Ph_2SiH_2 **2a** and allylic copper intermediates **16**, **17** and **18** could proceed via the four-membered cyclic transition states. Among the three transition states, **21-ts** (23.7 kcal/mol), which leads to the formation of (*E*)-**3a**, is found to be the most favorable one, thus, accounting for the domination of the (*E*)-allylsilanes product **3a** observed in the experiment.

(3) According to DFT calculations, σ -bond metathesis step are the reaction rate-limiting step and the stereoselectivity determining step, which is consistent to the KIE study ($k_H/k_D = 1.92$, Fig. R2).

(4) The *E*-**3a** is more stable than *Z*-**3a**, and we found that *Z*-**3a** did not isomerize to *E*-**3a** under our reaction conditions (Fig 5c of our revised manuscript).

Based on these results, we propose that stereoselectivity was kinetically controlled by the catalyst and we deleted the proposal “stereoselectivity is probably determined by the thermodynamic stability of the (*E*)-allylcopper intermediates” in the revised manuscript.

Fig. R1 Computational study. **a**) DFT calculations for the Cu-catalyzed hydrocupration of allene **1a**. **b**) Gibbs free energies required for the σ -bond metathesis step of Ph_2SiH_2 **2a** and three allylic copper intermediates **16**, **17** and **18**. Calculations were performed using Gaussian 16 at the M06-L/SDD-6-311++G(d,p)/SMD(THF)//B3LYP-D3(BJ)/LANL2DZ-6-31G(d) level of theory.

Fig. R2 KIE study

Significance

The paper makes an important addition to the field of hydrosilylation of terminal allenes. Authors report a simple method with a broad substrate scope to obtain both di- and trisubstituted allylsilanes with excellent regio- and stereoselectivity. I think these findings are of interest to a broad chemist community.

Our response: Thanks for the positive comments.

Data and methodology and Analytical Approach

In general, the data and findings have been presented to a very high standard. There are however several points the authors need to consider.

-¹⁹F NMRs are not reported for any of the fluorinated products.

Our response: Thanks for the comments. We have added the ¹⁹F NMRs for fluorinated products 3g, 3ab, 3m, 3t, 3aa, 3z, 5q, 5s, 5h, 5i, 5r.

-¹H NMR characterisation for compound **5t** is not clear. Signals integrating 0.33H or 1.4H (for example) don't make sense. Resonances corresponding to *E*- and *Z*- isomers should be listed separately when possible.

Our response: Thanks for the kind suggestions. We have listed resonances corresponding to *E*- and *Z*- isomers. *E* product: ¹H NMR (600 MHz, CDCl₃, 25 °C) δ 7.61–7.55 (m, 4H), 7.44–7.33 (m, 6H), 5.42 (dt, *J* = 15.3, 7.3 Hz, 1H), 5.30 (dd, *J* = 15.4, 6.9 Hz, 1H), 4.85 (t, *J* = 3.5 Hz, 1H), 2.05 (dd, *J* = 7.8, 3.5 Hz, 2H), 1.91–1.81 (m, 1H), 1.72–1.64 (m, 2H), 1.64–1.58 (m, 3H), 1.27–1.18 (m, 2H), 1.18–1.07 (m, 1H), 1.04–0.93 (m, 2H). ¹³C NMR (151 MHz, CDCl₃, 25 °C) δ 137.5, 135.4, 134.2, 129.7, 128.0, 121.9, 41.1, 33.4, 26.3, 26.2, 18.1. IR (ATR): ν 3067, 3011, 2922, 2847, 2120, 1446, 1155, 1114, 965, 797, 730, 700. HRMS (APCI⁺): (*m/z*) calcd for C₂₁H₂₇Si⁺ (*M*+H⁺), 307.1877; found, 307.1871. *Z* product: ¹H NMR (600 MHz, CDCl₃, 25 °C) δ 5.35–5.31 (m, 1H), 5.18–5.10 (m, 1H), 2.17–2.08 (m, 3H), other signals are overlapping with those of the *E*-isomer. *Z*- isomer is a known compound, and the characteristic spectroscopic data for this product matched the literature data (Nat. Commun. 8, 2258 (2017)).

-¹H NMR characterisation of **3a'** (deuterated **3a**) should be reported. If possible, include ²H NMR data.

Our response: Thanks for the kind suggestions. We have characterized **3a'** and the ²H NMR data has been added in the revised supporting information.

Suggested improvements

-Synthetical utility of the products could be proved for example by conducting coupling reactions.

Our response: Thanks for the helpful suggestions. Taking advantage of the *E*-configuration of compound **3a**, we achieved the intramolecular dehydrogenative C-Si bond formation via Ir catalyzed C-H activation, affording the five-membered silicon-containing compound **6** in 60% yield. Moreover, Allyl silanol **10** could be obtained in 90% yield under the Pd-catalyzed conditions. Compound **11** was then synthesized in 77% yield through the Hiyama-Denmark cross-coupling reaction between silanol **10** and PhI. These data have been added in the revised manuscript and supplementary information.

-Study of the KIE of the reaction could shed light into the mechanisms of the hydrosilylation reaction.

Our response: Thanks for the kind suggestion. We have conducted the KIE study and the value (1.92) indicates that Si-H bond might be involved in the rate-determining step.

Clarity and context

In general, the work is presented clearly. However, as stated above the authors need perhaps be a bit more precise on how stereoselectivity is controlled. Aspects of discussion of the prior literature (e.g. ‘*On the basis of the above experimental results and previous reports about Cu-H chemistry*’) may also be a little hard to follow for those without a detailed knowledge of this field.

Our response: Thanks for the kind suggestion. We have added the DFT calculation results and the KIE study in the revised manuscript to discuss the selectivity control. The discussion of prior literature has been added as follows: “It is known that CuH intermediates could be generated from the reaction of copper salts and silanes and CuH could add to olefins to generate organocopper intermediates.⁴²⁻⁴⁷ For the reaction of allenes with CuH, both terminal and internal double bonds of **1a** could participate in the hydrocupration.⁴⁶”

References

In general, references are appropriate. However, authors should consider include references about ‘*previous reports about Cu-H chemistry*’ in which the proposed mechanism is based (see above). A reference supporting synthetic value of allylic alcohols could be also added.

Our response: Thanks for the kind suggestions. Two papers on Cu-H catalyzed functionalization of allenes have been added as references 46 and 47 (Angew. Chem. Int. Ed. 55, 14077–14080 (2016); Chem. Sci. 11, 9115–9121 (2020).) Two reviews on the synthetic applications of allylic alcohols have been added as references 49 and 50 (Tetrahedron: Asymmetry 26, 405–495 (2015); Tetrahedron 111, 132732 (2022)). We are sorry for missing these papers in the last submission.

Reviewer #3 (Remarks to the Author):

This interesting manuscript describes the facile method for the synthesis of functionalized (E)-allylsilanes via copper-catalyzed regio- and stereoselective hydrosilylation of allenes.

There are not many examples of selective hydrosilylation of allenes because the reaction can potentially lead to many isomers.

The authors cite several examples of synthetic methods for obtaining (E)-allylsilanes via hydrosilylation of allenes but these examples are not without their drawbacks and limitations. Therefore, the reviewed work seems to be very interesting.

Unfortunately, a very similar paper on hydrosilylation of allenes has been published recently (<https://doi.org/10.1021/jacs.2c00260>).

The copper catalyst (Copper(I) thiophene-2-carboxylate) also proved to be an active and selective catalyst for the synthesis of functionalized (E)-allylsilanes in the presence of Xantphos. The investigations are very similar, and therefore, I do not see the possibility to publish this manuscript in this journal.

Our response: Thanks for the comments. The publication of the above mentioned paper on a high impact journal during the revision of our manuscript might support the novelty of our work. Moreover, the conditions of our work and the JACS paper are different. In addition, we have disclosed the substrate scope of the reactions between 1-aryl-1-alkyl allenes and diarylsilanes, but the JACS paper did not disclose these type of reactions.

However, some points should be addressed.

- Figure 2 and 3 are incorrect Ph_2SiH_2 instead of PhSiH_2 and $\text{Ph}(\text{Ar})\text{SiH}_2$ instead of PhSiH_2

Our response: Thanks for pointing out these mistakes. We have corrected these typos.

- PhSiH_3 and PhMeSiH_2 should also be tested

Our response: Thanks for the comments. Several examples with PhSiH_3 and PhMeSiH_2 as reagents have been added (compounds **3ac**, **5u**, **5w**).

3ac, 66% yield, 98:2 E/Z

5u, 72% yield, 98:2 E/Z

5w, 70% yield, 96:4 E/Z

- Are compounds 3a-3z and 3aa, 3ab the isomers shown in fig.2? There is a lack of evidence. This is not apparent from the NMR spectra. Far interaction spectra or X-ray structures are missing.

Our response: Thanks for the comments. The NOESY spectra of **3a**, **3ac** and **3g** confirmed the configuration of these compounds and the configurations of other compounds were assigned accordingly. These data have been added in the revised supplementary information.

- for the compounds from fig.3 only for a few examples ^1H NMR double bond coupling constants are shown.

Our response: Thanks for the comments. We corrected these mistakes in the revised supplementary information.

REVIEWERS' COMMENTS

Reviewer #1 (Remarks to the Author):

The revised manuscript is better than the previous one. No further revision is needed.

Reviewer #2 (Remarks to the Author):

Thanks to the authors for addressing the concerns of the reviewers. From my point of view, comments from the reviewers have been approached in an efficient and elegant way. Authors have been increased the initial broad scope of the reaction by including reactivity with mono-substituted alkyl allene (5t), and phenyl hydrosilane (3ac and 5u) and alkyl silanes (5v and 5w). They have clearly demonstrated the synthetic potential of the reported methodology by including two well-designed transformations of the products: intramolecular dehydrogenative C-Si bond formation (taking advantage of the (E)-configuration) and a Hiyama-Denmark coupling reaction. Regarding mechanistic investigation, the addition of DFT calculations and KIE studies strongly support the experimental data and conclusions drawn by the authors. Authors have also efficiently addressed concerns about compound characterisation and references. In my opinion, this simple and easy copper-catalysed method to prepare (E)-allylsilanes is suitable to be published in Nature Communications without any change.

Reviewer #3 (Remarks to the Author):

congratulations to the authors right now this paper looks very good.

One mistake:

in the description of the reaction conditions (figure2 and 3) should be R_2R_3SiH instead of Ph_2SiH_2

Reviewer #4 (Remarks to the Author):

In this work, Shen, Lan and co-workers reported an interesting Cu-catalyzed regio- and stereoselective hydrosilylation of terminal allenes. An excellent selectivity control is achieved, and a thorough mechanistic study is provided. I would mainly comment on the computational part as the editor required. I think the computational part is overall well executed. The mechanistic picture is clean and concise. This work is suitable for publication in Nature Communications in the following minor issues are properly addressed.

Minor issues:

1. Fig. 6 have too many different energy references. The energies are referenced to initial reactants, intermediate 16, 17 and 18 based on the circumstances. This is very misleading since all the species are in the same catalytic process and their energies can be compared. In Fig. 6a, only the initial 11 and 12 should be used as free energy reference. In Fig. 6b, 16, 17 and 18 are in fast equilibrium, and their relative free energies matter. If the authors want to emphasize the facile isomerization, simply add the barriers of 19-ts and 20-ts in Fig. 6b as well.

2. I cannot find any details of the phosphine ligand used in computations. Judging from the coordinate, I suppose it's Xantphos? Please clarify the details in the manuscript.

3. Last but most importantly, the authors didn't discuss anything on the competing transition states that determine the regio- and stereoselectivity. The calculated barriers of transition states (21-ts, 22-ts and 23-ts) agree well with the observations, but the authors' further elaborations based on their transition state model should be very helpful to the readers.

RESPONSE TO REVIEWER COMMENTS

Reviewer #1 (Remarks to the Author):

The revised manuscript is better than the previous one. No further revision is needed.

Our response: Thanks for the positive comments.

Reviewer #2 (Remarks to the Author):

Thanks to the authors for addressing the concerns of the reviewers. From my point of view, comments from the reviewers have been approached in an efficient and elegant way. Authors have been increased the initial broad scope of the reaction by including reactivity with mono-substituted alkyl allene (5t), and phenyl hydrosilane (3ac and 5u) and alkyl silanes (5v and 5w). They have clearly demonstrated the synthetic potential of the reported methodology by including two well-designed transformations of the products: intramolecular dehydrogenative C-Si bond formation (taking advantage of the (E)-configuration) and a Hiyama-Denmark coupling reaction. Regarding mechanistic investigation, the addition of DFT calculations and KIE studies strongly support the experimental data and conclusions drawn by the authors. Authors have also efficiently addressed concerns about compound characterisation and references. In my opinion, this simple and easy copper-catalysed method to prepare (E)-allylsilanes is suitable to be published in Nature Communications without any change.

Our response: Thanks for the positive comments.

Reviewer #3 (Remarks to the Author):

congratulations to the authors right now this paper looks very good.

One mistake:

in the description of the reaction conditions (figure2 and 3) should be R_2R_3SiH instead of Ph_2SiH_2

Our response: Thanks for the kind suggestion. We have corrected these typos.

Reviewer #4 (Remarks to the Author):

In this work, Shen, Lan and co-workers reported an interesting Cu-catalyzed regio- and stereoselective hydrosilylation of terminal allenes. An excellent selectivity control is achieved, and a thorough mechanistic study is provided. I would mainly comment on the computational part as the editor required. I think the computational part is overall well executed. The mechanistic picture is clean and concise. This work is suitable for publication in Nature Communications in the following minor issues are properly addressed.

Minor issues:

1. Fig. 6 have too many different energy references. The energies are referenced to initial reactants,

intermediate 16, 17 and 18 based on the circumstances. This is very misleading since all the species are in the same catalytic process and their energies can be compared. In Fig. 6a, only the initial 11 and 12 should be used as free energy reference. In Fig. 6b, 16, 17 and 18 are in fast equilibrium, and their relative free energies matter. If the authors want to emphasize the facile isomerization, simply add the barriers of 19-ts and 20-ts in Fig. 6b as well.

Our response: Thanks for the comments and helpful suggestions. In the revised Fig. 6, only the initial **1a** and **12** have been used as free energy reference and the barriers of **19-ts** and **20-ts** have been added.

Fig. 6 Computational study. **a)** DFT calculations for the Cu-catalyzed hydrocupration of allene **1a**. **b)** Gibbs free energies required for the σ -bond metathesis step of Ph_2SiH_2 **2a** and three allylic copper intermediates **16**, **17** and **18**. **c)** DFT calculations on the π - π interaction between the phenyl group of Xantphos and the phenyl group of the allene. Calculations were performed using Gaussian 16 at the M06-L/SDD-6-311+G(d,p)/SMD(THF)//B3LYP-D3(BJ)/LANL2DZ-6-31G(d) level of theory and the values are shown in kcal/mol.

2.I cannot find any details of the phosphine ligand used in computations. Judging from the coordinate, I suppose it's Xantphos? Please clarify the details in the manuscript.

Our response: Thanks for the kind suggestion. We add the description of Xantphos in revised Fig. 6.

3. Last but not most importantly, the authors didn't discuss anything on the competing transition states that determine the regio- and stereoselectivity. The calculated barriers of transition states (21-ts, 22-ts and 23-ts) agree well with the observations, but the authors' further elaborations based on their transition state model should be very helpful to the readers.

Our response: Thanks for the comments. According to the reviewer's suggestion, independent gradient model (IGM) analysis of the transition states **21-ts**, **22-ts** and **23-ts** was performed. As shown in revised Fig 6c, **21-ts** is 1.3 and 1.7 kcal/mol more favorable than **22-ts** and **23-ts**, which is consistent with the stereo- and regioselectivity observed in the experiment. The leading factor that differentiates the three competing transition states is likely to be the π - π interaction between the phenyl group of Xantphos ligand and the phenyl group of allylic compounds. This favorable π - π interaction stabilizes **21-ts** ($\Delta E_{\pi-\pi} = -2.1$ kcal/mol) by 0.7 and 2.0 kcal/mol relative to **22-ts** ($\Delta E_{\pi-\pi} = -1.4$ kcal/mol) and **23-ts** ($\Delta E_{\pi-\pi} = -0.1$ kcal/mol) based on the calculations of interacting fragments. The green oval represents the presence of interactions between the highlighted fragments. Therefore, our calculations indicate that the noncovalent π - π interaction is the determinant of stereo- and regioselectivity.